# Comparison of the Physical Fitness Profile of Muay Thai and Brazilian Jiu-Jitsu Athletes with Reference to Training Experience

**DOI:** 10.3390/ijerph19148451

**Published:** 2022-07-11

**Authors:** Wojciech Wąsacz, Łukasz Rydzik, Ibrahim Ouergui, Agnieszka Koteja, Dorota Ambroży, Tadeusz Ambroży, Pavel Ruzbarsky, Marian Rzepko

**Affiliations:** 1Independent Researcher, 33-100 Tarnów, Poland; poolbjj888@gmail.com; 2Institute of Sports Sciences, University of Physical Education, 31-571 Cracow, Poland; agnieszka.koteja@gmail.com (A.K.); dorota.ambrozy@awf.krakow.pl (D.A.); tadek@ambrozy.pl (T.A.); 3High Institute of Sport and Physical Education of Kef, University of Jendouba, Jendouba 8189, Tunisia; ouergui.brahim@yahoo.fr; 4Department of Sports Kinanthropology, Faculty of Sports, Universtiy of Presov, 080-01 Prešov, Slovakia; pavel.ruzbarsky@unipo.sk; 5College of Medical Sciences, Institute of Physical Culture Studies, University of Rzeszow, 35-959 Rzeszow, Poland; marianrzepko@poczta.onet.pl

**Keywords:** combat sports, motor fitness, training experience, Brazilian jiu-jitsu, Muay Thai

## Abstract

Background: In combat sports, successful competition and training require comprehensive motor fitness. The aim of this study was to diagnose the level of physical fitness and to determine the level of differences between athletes of combat sports characterized by stand-up fighting, such as Muay Thai; and ground fighting, such as Brazilian jiu-jitsu. Methods: The study examined and compared 30 participants divided into two equal groups: Muay Thai athletes (*n* = 15; age: 24.24 ± 3.24; body height: 174.91 ± 5.19; body weight: 77.56 ± 7.3), and Brazilian jiu-jitsu (BJJ) (*n* = 15; age: 22.82 ± 1.81; body height: 175.72 ± 7.03; body weight: 77.11 ± 8.12). Basic characteristics of the somatic build were measured. Selected manifestations of the motor potential of motor skills were also evaluated using selected tests from the EUROFIT test battery, the International Test of Physical Fitness, and computer tests of coordination skills. Relative strength and maximal anaerobic work (MAW) indices were calculated. The strength of the relationship between the effect of motor fitness and training experience was also assessed. Results: The athletes of both groups (Muay Thai and BJJ) presented similar levels of basic characteristics of the somatic build. Motor fitness in the tested groups showed significant differences between the athletes of these sports in static strength (*p* = 0.010), relative strength (*p* = 0.006), arm muscle strength in pull-ups (*p* = 0.035), and functional strength in bent arm hanging (*p* = 0.023). Higher levels of these components of motor fitness were found for the athletes in the BJJ athletes. In the Muay Thai group, significant very high strength of association was found between training experience and five strength tests. Furthermore, a significantly high strength of association was found in two tests. In the BJJ group, significant relationships with very high correlation were found between the variables in five strength tests. Conclusions: Brazilian jiu-jitsu athletes performed better in strength tests (static strength, relative strength, shoulder girdle strength, functional strength). High correlations between the training load and the level of physical fitness were found in flexibility and strength tests in BJJ athletes and most strength tests in Muay Thai athletes.

## 1. Introduction

Combat sports are characterized by diverse levels of competition. The fight can take place in stand-up positions (boxing, karate, kickboxing, taekwondo, and Muay Thai, which are striking sports) [1,2], as ground fighting (e.g., Brazilian jiu-jitsu—grappling sports), and in mixed positions (judo, jiu-jitsu, mixed martial arts) [1,3,4]. Structure of each fighting style requires specific physical preparation. Athletes often use exercises designed for another style in their preparation to increase the effectiveness and versatility of training [5]. The area of motor skills, which are individual and specific to an athlete, belongs to broad and very important issues of control of human sports activity [6]. Diagnosis and testing of physical fitness in sports are of great importance to the selection and control of progress in the training process [1,4,7]. In combat sports, sports career and training depend on comprehensive motor fitness including muscle strength, endurance, and speed, which results primarily from an above-average level of development of coordination and flexibility skills and strength–velocity and strength–endurance interactions that affect the comprehensive physical and mental development of athletes [2,5,7,8]. In recent years, several authors representing various research centers have focused on investigating the motor skills in combat sports such as kickboxing [2,9,10], jiu-jitsu [11], karate [12], Olympic taekwondo [13,14], judo [15], and wrestling [16,17]. Researchers are increasingly attempting to identify indicators of technical and tactical skills in various combat sports [8,18,19], the degree of physiological reactions during the fight [20,21,22,23], and biomechanics of movement [24,25]. The literature review, however, shows a lack of studies comparing two separate combat sports characterized by different rules and specifics of the fights that are gaining popularity in the international arena. Therefore, in this study it was decided to analyze Muay Thai and Brazilian jiu-jitsu athletes, with a combination of these sports resembling competition similar to mixed martial arts (MMA) and which are often referred to as the base combat sports in the presented form of competition. According to the authors of the study, the combination of skills developed in the two discussed combat sports may constitute an excellent motor background for MMA fighting. It is worth noting that in addition to physical preparation, the performance of the athlete can be determined by the level of technical and tactical skills [26,27]. Technical perfection and tactical capabilities are acquired with training and competitive experience [28].

Muay Thai (Thai boxing) is a ‘cultural export’ and the most popular sport in Thailand. The sport has a great influence on the world of combat sports and commercial trends [29]. Thai boxing is termed an ‘eight-limb’ martial art, with practitioners using eight points of contact with the opponent (i.e., fists, elbows, knees, and feet). Furthermore, it includes grappling techniques (involving takedowns and clinch) [1,29]. Brazilian jiu-jitsu (BJJ) is the flagship ground fighting sport developed in Brazil. It is a hybrid martial art derived from traditional jiu-jitsu, judo, and wrestling [30,31]. Specialized movement techniques are used in the fights, including limb levers, chokes, rolls, throws, and takedowns [32]. The sport is also one of the pillars on which all-style hand-to-hand combat known as mixed martial arts (MMA) is based [33]. Considering the above, the aim of the present study was to diagnose the level of physical fitness and to determine the level of differences between athletes of combat sports characterized by stand-up fighting such as Muay Thai, and ground fighting, such as Brazilian jiu-jitsu. An additional aim was to determine the relationship between training experience and physical fitness in the two fighting styles studied.

## 2. Materials and Methods

### 2.1. Participants

The evaluation of motor fitness and the basic measurements of the somatic build of athletes was conducted in the KS Legion Team Tarnów sports club in Tarnów, Poland. The measurements were carried out between 15 and 23 June 2021. The observation was based on the sample size determined using G*Power software 3.1.9.4. Two groups (15 athletes each) were formed based on the calculations: the Muay Thai athletes (group 1; *n* = 15; mean ± SD age: 24.24 ± 3.24 years) and Brazilian jiu-jitsu (BJJ) practitioners (group 2; *n* = 15; mean ± SD 22.82 ± 1.81 years) (Figure 1).

The training experience of the athletes of both sections of the club ranged from 3 to 6 years. Mean training experience was 4.6 years ±1.55 in the Muay Thai group and 4.4 years ±1.55 in the BJJ group. Information about chronological age and training experience was obtained by means of a diagnostic survey using the questionnaire technique. The athletes studied were actively participating in competitions at international, national, and local levels, and some of them had achieved significant sports results. The research was approved by the bioethics committee at the Regional Medical Chamber (no. 287/KBL/OIL/2020).

Participants did not have any medical contraindications or acute or chronic injuries and were not following body mass reduction programs during the entire experiment. The inclusion and exclusion criteria are shown in Table 1. They were also asked to refrain from any strenuous effort 24 h before the testing sessions. The study was conducted in accordance with the Declaration of Helsinki for human experimentation and the protocol was fully approved by the local research ethics committee before the start of the study. All athletes gave written informed consent after a detailed explanation of the aims, benefits, and potential risks involved in the research.

### 2.2. Testing Procedures

Measurements took place in a sports hall where sports training of the tested athletes is conducted on a daily basis. All measurements were performed at the same time of day (i.e., between 5:00 p.m. and 8:00 p.m.) to avoid any diurnal variation of the performance, at specially designed test stands during 10 standard training sessions, with each training session lasting 90 min. Only one of the observed groups was present in the testing room at a time. The measurements for each group were carried out during five consecutive training sessions. The participants were thoroughly familiarized with all testing procedures.

### 2.3. Anthropometric Measurements

Selected characteristics of the somatic build were measured according to the recommendations used in anthropometry, i.e., body height in centimeters (cm) with an accuracy of 1 cm, body weight in kilograms (kg) with an accuracy of 0.1 kg, and body fat percentage. Prior to the somatic measurements, participants were asked to remove their clothes and remain in their underwear only. The body height was determined as the distance between the vertex (v) and basis (B) points, i.e., the highest point of the head in the Frankfurt plane and the ground plane on which the tested person stood in an upright position with arms along the body. During the measurement, the participant’s heels remained joined and the feet were slightly apart. The v-B distance was measured using an A213 anthropometer. An approved electronic scale TANITA TBF-538 was used to measure body weight. The value of body weight in kilograms and the percentage of body fat were read from the scale and recorded.

### 2.4. Motor Fitness Assessements

Selected tests from the International Physical Fitness Test and EUROFIT battery standardized physical fitness tests were used to assess the level of motor fitness. All tests used were verified for accuracy and reliability [4,10,34,35]. Each time before the measurements, both groups participated in a standard 15-min warm-up session consisting of exercises to prepare the body for physical effort. Exercises were conducted in accordance with the principle of formative exercises and involved static and dynamic movements of the arm, trunk, abdomen, back, and legs. The assessment of motor skills was carried out each time in the order established before the measurements.

Equilibrium posture (static balance): the subject stands up on a bar with a length of 50 cm, height of 4 cm, and width of 3 cm. The subject then holds the free leg bent at the knee from behind the foot. The subject’s task is to maintain balance for as long as possible. The measurement ends when the subject loses his or her balance, i.e., lets go of the leg or touches the ground. The subject is allowed to perform one pre-trial prior to the measurement. The time is measured to the nearest 0.01 s [34].Simple reaction time: testing takes place at the computer keyboard. Active Keys: “Enter”, on the right for the right hand and “1”, on the left for the left hand. The subject places his or her hand next to the keyboard so that it rests comfortably on the table, with their thumb on the active key. When a bright square appears in the center of the screen, the subject is supposed to press the active key as soon as possible. In the test, this process is repeated irregularly as 11 pulses. The faster the response, the better the outcome. The examiner demonstrates the task, then gives instructions and explanations, and the test subject immediately performs a pre-trial of 5 pulses and then proceeds to the main test of 11 pulses [34].Trunk flexibility: the test is performed as a sit-and-reach movement, with the range of motion measured in cm, below the feet level. In a seated position, the subject reaches the arms forward as far as they can. The subject, in a straddle sitting position, reaches forward with the hands as far as possible by sliding the ruler on the surface of the box with a previously prepared scale. The better of the two results is recorded. If the participant reaches 10 cm beyond the toes, they receive a score of 10. The box that is used is 40 cm long, 45 cm wide, and 35 cm high, and a 65 cm long graduated box top protrudes 25 cm over the side wall that marks the width of the box and is used as a feet rest; the box top is fixed in such a way that the graduation mark drawn on it indicates 50 in the place where feet touch the surface of the box; a 30 cm long ruler placed loosely on the surface of the box perpendicularly to its longitudinal axis and used for moving with hands while performing a forward reach [2].Static handgrip strength: the subject stands with a small straddle, with the dynamometer held tightly in the fingers, the arm is positioned along the body so that the hand does not touch the body; the subject performs a short grip on the dynamometer with maximum force, with the other arm along the body. The better result of the two hand tests is recorded to the nearest 1 kg [2].Relative handgrip strength: relative strength is a strength index that represents the ratio of absolute force generated by the muscles to the total body mass or lean body mass (LBM). Dynamometric measurement of handgrip strength was expressed in relative units. Relative strength was calculated as the quotient of the result of the measurement performed with a handgrip tester (result in kG) and the body weight (kg) of the subject. This measure gives an objective and accurate picture of the characteristics of real muscle strength, which is very important in sports limited by weight categories (Szopa et al., 1996), including Muay Thai and BJJ [35].
$$SW=\frac{dynamometer\;result\;\left[\mathrm{kG}\right]}{body\;mass\;\left[\mathrm{kg}\right]}\;$$Long jump (explosive power): the subject stands with the feet slightly apart in front of the starting line, bends the knees, and moves the arms backward at the same time, and then he or she performs the arm swing and jumps as far as they can; the landing occurs on both feet while maintaining the upright position; the test is performed twice. The longest of the two jumps measured to the closest mark left by the participant’s heel is recorded, with an accuracy of 1 cm. A tape measure, a hard surface, and two gymnastic mattresses connected lengthwise are used [9].Maximal anaerobic alactic power (MAP): is the ability to perform maximal work as fast as possible to assess the level of speed and strength abilities. In indirect tests, it is recommended to measure maximum anaerobic work (MAW), which is an approximate measure of MAP. The MAW was calculated from the standing long jump test results as a product of the jump measurement result (m), the subject’s body mass (kg), and gravitational acceleration [35].
$$MW=jump\;height\;\left[m\right]\times body\;mass\;\left[\mathrm{kg}\right]\times 9.81\;\left[\frac{m}{s^{2}}\right]\;$$Shuttle run 10 × 5 m: The participant runs on a signal to the second line 5 m away, crosses it with both feet, and comes back. They run a distance of 5 m 10 times. The time of the shuttle run is measured and rounded to a decimal place of a second [2].Sit-ups: Evaluation of abdominal strength: the tested person lies on the mattress with feet 30 cm apart and knees bent. Hands intertwined, resting on the nape of the neck, feet hooked to the ladder so that they remain in contact with the ground. At the signal, the participant sits up to touch the knees with elbows and then returns to the starting position. The exercise duration is 30 s [10].Leg strength, barbell squats with 50%BM: Leg muscle strength was assessed with a squat performed with 50% body mass. The subject begins in a standing position with feet hip-width apart. The subject holds a barbell with a set weight on his or her shoulders behind their head. The task is to perform as many squats with a barbell as possible. The test is performed once [34].Arm strength, pull-ups: evaluation of the strength of the shoulder girdle based on the number of repetitions: the subject catches the bar using a pronated grip and hangs; at the signal, the subject bends arms in elbows and pulls the body up so high that the chin is above the bar and then, without rest, returns to a simple hanging; the exercise is repeated as many times as possible without rest; the result is the number of complete pull-ups (chin over the bar) [34].Push-ups (muscle strength): the subject performs a front support position with arms shoulder-width apart. At the signal, the subject performs push-ups (to the level of the ground) with full arm extensions (as fast as possible with as many repetitions as possible in 30 s) [34].Flat bench press with 50%BM: The subject begins the test by lying on a flat bench designed for bench press exercises. Then he or she grips a barbell with arms shoulder-width apart with a set weight. The test consists of performing as many bench press repetitions as possible by flexion (up to the chest level) and extension of the shoulder and elbow joints. The test is performed once [34].Back extension (strength endurance of the back muscles): The subject is lying prone on a mattress. Hands are intertwined, resting on the neck. At the signal, the subject, from the lying prone position, bends the trunk backward by contracting the muscles of the back, together with the legs (the body forms an arch), and then returns to the starting position (lying prone) as fast as possible, and repeats this sequence, also as fast as possible, with as many repetitions as possible within 30 s. The examiner counts the number of repetitions performed in 30 s. For example, 18 correctly executed bends, results in 18. The test is performed once [34].Hanging with arms bent (functional strength/muscle isometrics test): the test consists of hanging on a bar with arms flexed at the elbows so that the chin is above the bar (the exact angle is not specified, it is important that the chin is above the bar). A stopwatch is started at the moment when independent hanging begins. Time measurement continues as long as the subject’s eyes are above the bar. The test is performed once. The hanging time is measured to the nearest 0.1 s [35].

### 2.5. Statistical Analyses

Statistical analysis was performed using Statistica software (Statsoft, Kraków, Poland) version 13.3. Basic descriptive statistics were calculated (arithmetic mean, median, standard deviation, minimum and maximum values, range of variation, and coefficient of variation). The analysis was performed on logarithmic data, and the assumption of conformity of the distribution with normal distribution was met as verified by the Shapiro–Wilk test. Differences between groups were evaluated using the Student’s *t*-test for independent variables in both cases. Pearson’s linear correlation was used to determine relationships between measured parameters. The level of statistical significance was set at *p* < 0.05. Correlations were classified as weak at 0.2–0.4, moderate at 0.4–0.7, strong at 0.7–0.9, and very strong at >0.0 [36].

## 3. Results

The measurement of basic somatic physical characteristics (Table 2) revealed no significant differences between athletes in the two sports. Compared to the individuals from the BJJ group, Muay Thai athletes had slightly higher body weight, BMI, and body fat percentage; whereas BJJ athletes were characterized by slightly higher body height.

Analysis of the results presented in Table 3 leads to the conclusion that in terms of the level of motor fitness and its intergroup variation, the athletes from the discussed groups differed in selected motor skills. The comparative analysis (Table 3) revealed that in the case of static strength, relative strength, arm muscle strength (pull-ups on a bar), and the time of hanging with arms bent in an isometric contraction, the athletes from the BJJ group had significantly higher levels compared to Muay Thai athletes. For abdominal strength endurance, push-ups, back muscle strength, barbell squats, and barbell bench press, higher but insignificant differences were observed in the BJJ group. There were no differences between the groups in static balance, simple reaction time to a visual stimulus, trunk flexibility, explosive strength of lower limbs, and the resulting index of maximum anaerobic work and agility test results.

The training experience of Muay Thai athletes was 4.6 ± 1.55, while that of BJJ athletes was 4.4 ± 1.55 (*p* > 0.05). In the Muay Thai group, a statistically significant effect of training experience was observed in seven strength tests. Very high correlations were found in five of them, i.e., barbell squats, pull-ups, forward bending, back extensions, and hanging with bent arms. On the other hand, the other two tests (standing long jump and flat bench press) found high strength. Analysis of the correlations in BJJ athletes between the mentioned variables revealed significant statistical correlations for 11 motor skills. Relationships with very high correlation were recorded in five strength tests: sit-ups, barbell squats, flat bench press, back extensions, and hanging with arms bent. High correlations were also found in four strength tests (static strength, relative strength, long jump, and pull-ups) and one flexibility test (forward trunk bending). For the measurement of simple reaction time, a high negative correlation was found (Table 4).

## 4. Discussion

The aim of this study was to diagnose the level of physical fitness and to determine the level of differences between athletes of combat sports characterized by stand-up fighting, such as Muay Thai; and ground fighting, such as Brazilian jiu-jitsu. The results of the present study showed a diversified picture of the motor profile of individual combat sports athletes in the groups studied. Therefore, it seems reasonable to assume that, among others, an environmental factor in the form of the source style (combat sport) of the athlete influenced the dominance of a group in individual motor fitness tests. Analysis of the level of motor fitness in the present study revealed that both Muay Thai and BJJ athletes differed in static strength, relative strength, arm muscle strength, and the time of hanging with bent arms in isometric contraction as an expression of functional strength. Higher significance levels of these fitness components were found in BJJ athletes compared to the entire study population. The following tests also showed favorable results in BJJ athletes: sit-ups, push-ups, back extensions, barbell squats, pull-ups, and flat bench press. BJJ athletes outperformed the control group in strength and power endurance tests. The results are similar to those reported in a study [37] conducted in 2019 among MMA and BJJ athletes of different ages and training experience. Specifically, BJJ fighters showed significantly higher values for static strength, relative strength, and resistance of abdominal muscles to fatigue compared to MMA fighters [37]. The role and importance of the handgrip strength of the fighting athletes—especially in grappling sports such as judo, wrestling, and Ne-Waza jiu-jitsu—is very well supported by scientific evidence, with special emphasis on the gi fighting style. The results correspond positively with the measurements of sports jiu-jitsu athletes. Ambroży [38] showed that jiu-jitsu competitors who perform a lot of gripping actions presented a strength advantage in this aspect compared to karatekas, who were characterized by significantly less frequent gripping, with the prevalence of punches and kicks, similarly to Muay Thai competitors. A high level of development of this motor aspect among grapplers and wrestlers compared to strikers was also shown by Adamczyk et al. [39]. Sanchez et al. [40] showed an analogy in this matter, illustrating additionally the utilitarian aspect of this form of movement in athletes with proper technical preparation. They found that techniques in grappling combat sports are prevalent in self-defense training as they are a very important element in situations when it is necessary to counteract an attack [40]. In a study comparing judo and jiu-jitsu athletes, an advantage of the former in endurance and strength tests was observed. This could be explained by the effect of randori training fights during which endurance and strength abilities determine gaining an advantage over the opponent [41]. In its training process, BJJ also uses a form of task fighting called ‘kakari geiko’ (meaning ‘stress training’ or ‘pressure training’). In these movement tasks, all-round strength is strongly developed and perfected in every aspect and equally often ensures an advantage over the opponent. Although the above reports concern other combat sports, and BJJ originates from traditional jiu-jitsu and judo and belongs to grappling sports, its specificity is inextricably linked with these sports. Based on thermographic analysis of muscle work, it was found that BJJ is a very specific sport as it was shown that large muscle groups (i.e., quadriceps, gluteus maximus, latissimus dorsi) are involved during training, with abdominal muscles and back extensors mostly used. This is due to the frequent and intense use of hip work during the performance of various techniques in this sport [42]. The review of reports by previous authors seems to explain the predominance of strength and strength endurance in this group. Therefore, it can be concluded that the specific BJJ training leads to a significant improvement in the level of competence in this area. The specificity of this training involves the regular development of special movement skills which are based on a high level of multidimensional strength as part of the motor potential.

Analysis of the level and intergroup variation of other aspects of motor fitness revealed an insignificantly higher performance of Muay Thai athletes in terms of motor skills such as speed, coordination, flexibility, and power. In the tests that evaluated static balance, simple reaction time to a visual stimulus, trunk flexibility, explosive strength of lower limbs, the resulting index of maximum anaerobic work, and agility, better results were found in this group. Ambroży [43] showed that, compared to sports jiu-jitsu (specificity similar to BJJ), karate training has a more beneficial effect on the level of speed, leg and abdominal muscle strength, and flexibility. Furthermore, this author stressed that during training activities, karate athletes put more emphasis on flexibility, which helps them gain speed and precision when performing kicks and punches [43]. The results of the present study confirm these findings in terms of speed, flexibility, and explosive strength of the lower limbs. High agility competence allows the athlete to move smoothly in a ring, which increases the effectiveness of attacks (one can surprise the opponent with agility, change of pace, and anticipation of the attack) and improves defensive actions (dodges, blocks). Speed and coordination are basic abilities used in kickboxing (a sport very similar in specificity to Muay Thai) and lie at the basis of proper timing, which means using a technique at the right time [2]. Most likely, the specificity of training also gives Muay Thai practitioners an advantage in this case. In order to perform technical actions using different types of kicks and punches, stand-up fighters must develop a high level of speed, flexibility, coordination, agility, and static and dynamic balance, which may explain their performance in the above-mentioned tests.

The results may indicate different recruitment methods or different scopes of methods of motor learning and motor training. In BJJ athletes, training is much more strength-oriented; while in Muay Thai, the development of speed, flexibility, coordination, or agility hybrids seems to be a higher priority. Turner [44] reported that most athletes in the latter sport are reluctant to undergo strength training due to their concerns about the loss of flexibility and weight gain. In general, motor fitness in the studied populations did not differ significantly from each other except for specific aspects of strength. It is highly likely that sports training in both combat sports groups has a beneficial effect on the comprehensive motor preparation of the athletes.

Based on the correlation coefficients obtained in the tests, the greatest and statistically significant relationship with training experience was found for strength tests (Muay Thai: r = 0.65–0.85, BJJ: r = 0.57–0.87). The effect of training experience was the strongest in five strength tests for both tested groups, with high correlations. A high correlation was also found in the Muay Thai group for two tests and in the BJJ group for six tests. The results of the study indicate that in Brazilian jiu-jitsu, training experience plays a significant role in the development of strength, which may result from the specificity of the sport because most of the fights occur in low positions (ground fighting). Such exercise requires strength and strength endurance of the limbs and abdominal muscles [45]. In Muay Thai, it is essential to develop leg and arm strength, which allows the athlete to perform punches, kicks, and clinching. Therefore, Muay Thai training is oriented towards the development of limb strength, which increases with training experience as evidenced by statistically significant correlations (r = 0.67 *p* < 0.001).

In conclusion, of the 15 manifestations of motor fitness measured in the study, the BJJ group had the best results in nine, four of which showed significant intergroup differences. BJJ athletes presented a motor fitness profile characterized by higher strength and power endurance. The Muay Thai group insignificantly outperformed them in six aspects. The motor profile of the Muay Thai boxing athletes showed higher levels of speed, coordination, and flexibility. Such results show the presence of elements of physical fitness specific to the fighting styles studied. It can be assumed that the specific training in these sports leads to the development of certain motor skills.

## 5. Conclusions

1. There are statistically significant differences in static handgrip strength, relative strength, arm strength, and hanging with arms bent between Muay Thai and Brazilian jiu-jitsu (BJJ) athletes.

2. Significant correlations were observed between training experience and the development of strength motor skills in both studied groups.

### Practical Applications

The results of this study can be used by coaches as guidelines for developing strength and conditioning programs for their athletes in the preparation for competitions in combat sports. This seems to be especially beneficial in the preparation for MMA tournaments, when specific deficiencies caused by the athlete’s focus on their basic combat sports should be addressed.

## Figures and Tables

**Figure 1 ijerph-19-08451-f001:**
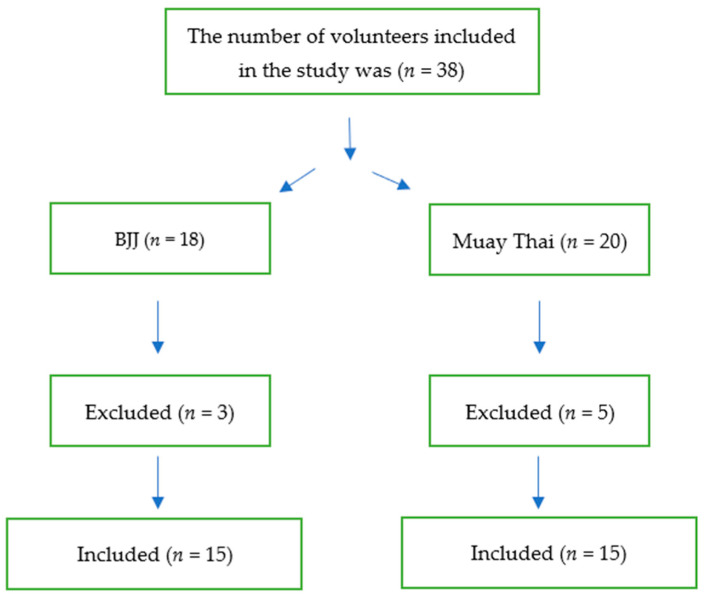
Participants’ flow diagram.

**Table 1 ijerph-19-08451-t001:** Inclusion and exclusion criteria.

Inclusion	Exclusion
Training experience of at least 3 years	Training experience of less than 3 years
Active participation in competitions	No participation in competitions
Male	Female
+ 18 to 30	Children
Good health status	Diseases of the joints and cardiovascular diseases

**Table 2 ijerph-19-08451-t002:** Somatic build of Muay Thai and BJJ athletes (*n* = 30).

Parameter	Group	$\tilde{X}$	Me	sd	Min	Max	R	V	*p*
Body height (cm)	Muay Thai	174.91	175	5.19	164.5	183.7	19.2	2.97	0.721
	BJJ	175.72	173.8	7.03	161.5	187.4	25.9	4.00	
Body weight (kg)	Muay Thai	77.56	78.7	7.3	61.2	89.7	28.5	9.41	0.873
	BJJ	77.11	79	8.12	62.1	92.8	30.7	10.54	
BMI	Muay Thai	25.36	25.99	2.27	20.98	27.7	6.73	8.93	0.255
	BJJ	24.97	25.09	2.2	21.13	29.02	7.89	8.81	
Fat content (%)	Muay Thai	20.05	20.5	3.92	12.8	25.2	12.4	19.53	0.278
	BJJ	18.44	19.5	4.03	11	24.5	13.5	21.87	

$\tilde{X}$—arithmetic mean; Me—median; sd—standard deviation; min—minimum value; max—maximum value; R—range; V—variance.

**Table 3 ijerph-19-08451-t003:** Motor fitness of studied Muay Thai and BJJ athletes (*n* = 30).

Parameter	Group	$\tilde{X}$	Me	sd	Min	Max	R	V	*p*
Equilibrium posture (static balance) (s)	Muay Thai	11.38	10.01	4.69	4.92	18.44	13.52	41.19	0.139
	BJJ	10.95	8.83	8.09	3.62	35.34	31.72	73.87	
Simple reaction time (s)	Muay Thai	0.229	0.238	0.029	0.185	0.291	0.106	12.844	0.148
	BJJ	0.242	0.245	0.019	0.218	0.286	0.068	8.029	
Trunk flexibility (cm)	Muay Thai	27.87	29	6.45	16	38	22	23.13	0.766
	BJJ	27.13	29	6.97	16	40	24	25.68	
Static handgrip strength (kG)	Muay Thai	44.6	42.4	8.47	31.7	58.6	26.9	18.99	**0.002**
	BJJ	54.58	56.3	11.25	33.6	71.5	37.9	20.62	
Relative handgrip strength	Muay Thai	0.58	0.55	0.11	0.45	0.79	0.34	18.45	**0.002**
	BJJ	0.71	0.70	0.14	0.43	0.91	0.48	19.9	
Long jump (explosive power) (cm)	Muay Thai	237.87	235	13.59	221	263	42	5.71	0.674
	BJJ	235.47	228	17.15	211	263	52	7.29	
MAW (J)	Muay Thai	1805.47	1800.42	150.91	1428.89	2030.48	601.6	8.36	0.758
	BJJ	1783.06	1857.98	235.05	1285.41	2056.27	770.86	13.18	
Shuttle run 10 × 5 m (s)	Muay Thai	20.66	20.21	2.05	18.05	25.58	7.53	9.94	0.685
	BJJ	20.91	20.56	1.16	19.44	22.58	3.14	5.54	
Sit-ups (*n*)	Muay Thai	30.67	30	3.11	26	36	10	10.14	0.067
	BJJ	33.67	34	5.25	23	42	19	15.58	
Leg strength, barbell squats with 50%BM (*n*)	Muay Thai	38.2	39	5.16	26	45	19	13.5	0.418
	BJJ	40	38	6.75	31	52	21	16.88	
Arm strength, pull-ups (*n*)	Muay Thai	9.93	10	3.24	5	18	13	32.61	**0.035**
	BJJ	13.6	13	5.57	6	26	20	40.92	
Push-ups (*n-*30 s)	Muay Thai	40.27	38	10.1	24	53	29	25.07	0.165
	BJJ	45.13	45	8.57	30	57	27	18.98	
Flat bench press with 50%BM (*n*)	Muay Thai	35.07	38	6.41	25	43	18	18.27	0.117
	BJJ	38.4	39	4.76	30	45	15	12.4	
Back extension (*n-*30 s)	Muay Thai	45.93	46	4.79	39	53	14	10.42	0.410
	BJJ	47.73	49	6.83	36	58	22	14.31	
Hanging with arms bent (s)	Muay Thai	32.11	32.24	17.13	9.53	63.49	53.96	53.35	**0.023**
	BJJ	46.97	49.56	16.78	18.02	72.34	54.32	35.73	

$\tilde{X}$—arithmetic mean; Me—median; sd—standard deviation; min—minimum value; max—maximum value; R—range; V—variance, statistically significant values are shown in bold.

**Table 4 ijerph-19-08451-t004:** Correlation coefficients of motor tests results and training experience (*n* = 30).

Motor Performance Tests	Muay Thai	Brazilian Jiu-Jitsu
Equilibrium posture (static balance) (s)	r = 0 *p* > 0.05	r = 0.12 *p* > 0.05
Simple reaction time (s)	r = 0.06 *p* > 0.05	r = −0.56 *p* < 0.05
Trunk flexibility (cm)	r = 0.06 *p* > 0.05	r = 0.76 *p* < 0.01
Static handgrip strength (kg)	r = 0.06 *p* > 0.05	r = 0.80 *p* < 0.01
Relative handgrip strength	r = 0.25 *p* > 0.05	r = 0.71 *p* < 0.01
Long jump (explosive power) (cm)	r = 0.67 *p* < 0.01	r = 0.65 *p* < 0.05
MPA (J)	r = −0.12 *p* > 0.05	r = 0.49 *p* > 0.05
Shuttle run 10 × 5 m (s)	r = 0.46 *p* > 0.05	r = −0.28 *p* > 0.05
Sit-ups (*n*)	r = 0.20 *p* > 0.05	r = 0.76 *p* < 0.01
Leg strength, barbell squats with 50%BM (*n*)	r = 0.85 *p* < 0.01	r = 0.87 *p* < 0.01
Arm strength, pull-ups (*n*)	r = 0.79 *p* < 0.01	r = 0.57 *p* < 0.05
Push-ups (*n-*30 s)	r = 0.82 *p* < 0.01	r = 0.17 *p* > 0.05
Flat bench press 50%BM (*n*)	r = 0.65 *p* < 0.05	r = 0.87 *p* < 0.01
Back extension (*n-*30 s)	r = 0.78 *p* < 0.01	r = 0.87 *p* < 0.01
Bent arm hanging (s)	r = 0.74 *p* < 0.01	r = 0.87 *p* < 0.01

r—Pearson’s correlation; *p*—significance of differences.

## Data Availability

All data are presented in the study.

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
