# Peer review of "Comparison of the Physical Fitness Profile of Muay Thai and Brazilian Jiu-Jitsu Athletes with Reference to Training Experience"

_ijerph, 2022, doi:10.3390/ijerph19148451_

Round 1

Reviewer 1 Report

Introduction

The introduction is well written and provides the rationale for the present study.

Materials and Methods

The motor fitness tests need further detail. These should be described in enough detail that another researcher could perfectly replicate what was performed in the present study. Please provide the details in the questions below:

What was the visual computerized test and how was it implemented?

What were the methods for the trunk flexibility test?

How specifically was the handgrip dynamometer test employed?

What were the instructions for the standing long jump test?

How was rate of muscle recruitment measured? Measuring rate of muscle recruitment infers that electromyography was used. If so provide these details, if not then eliminate this phrase. Provide the details for the methods for the 10 x 5 m shuttle run. 

What was the sit up test and push up test for muscular endurance?

Why was 50% of BM used for the squat test? I question this and the bench press as assessments of strength due to the relatively light intensity that 50% body mass is considered. Provide references that this has been used as an assessment of strength.

How is the pronated hanging on a bar a measure of functional strength that is not covered by the previous tests? What was the angle of the elbow joint during this test?

Statistical Analyses

How were the magnitudes of the correlation coefficients classified? Meaning, what was considered very high, high, moderate, low, and negligible?

Why were both Spearman's rank correlation and Pearson's correlation performed? Spearman is only appropriate if the data are not normally distributed.

Specify which tests were normally distributed and not normally distributed either here or in the results section.

Discussion

The article title is "Relationship between training experience and motor fitness profile in athletes of modern combat sports: Muay Thai and Brazilian jiu-jitsu." However, the smallest paragraph of the discussion is the paragraph that discusses the results of the correlation analyses (relationships). Is the primary focus on the relationships within these sports or the differences? If it is more about the relationships between training experience and motor fitness profile then this should be the primary focus and main bulk of the discussion section.

The final two paragraphs both begin with "In conclusion..." Combine these paragraphs as your conclusion paragraph for the discussion section.

Author Response

Dear Reviewer,

Thank you very much for your time and valuable comments, which all have been considered and incorporated. The detailed list of responses is given below. We hope that the modifications and explanation will be acceptable for you.

Yours sincerely,

Rydzik, corresponding author

Introduction

The introduction is well written and provides the rationale for the present study.

A: Thank you, as suggested by the second reviewer we have improved it again

Materials and Methods

The motor fitness tests need further detail. These should be described in enough detail that another researcher could perfectly replicate what was performed in the present study. Please provide the details in the questions below:

What was the visual computerized test and how was it implemented?

A: This has been corrected

What were the methods for the trunk flexibility test?

A: This has been corrected

How specifically was the handgrip dynamometer test employed?

A: This has been corrected

What were the instructions for the standing long jump test?

A: This has been corrected

How was rate of muscle recruitment measured? Measuring rate of muscle recruitment infers that electromyography was used. If so provide these details, if not then eliminate this phrase. Provide the details for the methods for the 10 x 5 m shuttle run. 

A: This has been corrected

What was the sit up test and push up test for muscular endurance?

A: This has been corrected

Why was 50% of BM used for the squat test? I question this and the bench press as assessments of strength due to the relatively light intensity that 50% body mass is considered. Provide references that this has been used as an assessment of strength.

A: This has been corrected

How is the pronated hanging on a bar a measure of functional strength that is not covered by the previous tests? What was the angle of the elbow joint during this test?

A: This has been corrected

Statistical Analyses

How were the magnitudes of the correlation coefficients classified? Meaning, what was considered very high, high, moderate, low, and negligible?

A: Added information

Why were both Spearman's rank correlation and Pearson's correlation performed? Spearman is only appropriate if the data are not normally distributed.

A: This was a transcription error

Specify which tests were normally distributed and not normally distributed either here or in the results section.

A: This has been corrected

Discussion

The article title is "Relationship between training experience and motor fitness profile in athletes of modern combat sports: Muay Thai and Brazilian jiu-jitsu." However, the smallest paragraph of the discussion is the paragraph that discusses the results of the correlation analyses (relationships). Is the primary focus on the relationships within these sports or the differences? If it is more about the relationships between training experience and motor fitness profile then this should be the primary focus and main bulk of the discussion section.

A: This has been corrected

The final two paragraphs both begin with "In conclusion..." Combine these paragraphs as your conclusion paragraph for the discussion section.

A: This has been corrected

Reviewer 2 Report

Abstract:

Aim of the study is a general statement – It should be specific

The conclusion is not in line with the aim of the study

Introduction:

Line 45- Notable to understand what the authors wanted to convey.

Line 47 – define comprehensive motor fitness

The introduction does not contain the proper background of the study. There are a number of sentences written without any coherence.

Line 72 – Why this sentence in the introduction section.

What is the rationale of the study? Why did the authors want to compare two sports?

Methodology:

How was the sample size calculated?

Add ethical approval details.

Line 105 and 107 – was there any training session for the participants

How the participant was selected? How many volunteered? How many were excluded. Include a flow diagram

Line 119 – provide details of the instrument

The procedure has to be written in detail including the procedure for each test

Reliability and validity of the tests should be added to the methodology

Li e 187 – cause-effect cannot be established in a cross-sectional study

The correlation coefficient is not available in results and table

Where is the training experience data?

Discussion

Line 202 – is this the aim of the study?

One of the most important part of your study is the correlation between training experience and fitness profile. This part is not discussed in the discussion section. Only one small paragraph with some results is available in the discussion regarding this. Even training experience data is not available in the results

Line 280 – where is the correlation coefficient?

Line 293- how you can conclude with this statement

Line 297- wrong statement

Conclusion not in line with the aim 0f the study 

Author Response

Dear Reviewer,

Thank you very much for your time and valuable comments, which all have been considered and incorporated. The detailed list of responses is given below. We hope that the modifications and explanation will be acceptable for you.

Yours sincerely,

Rydzik, corresponding author

Abstract:

Aim of the study is a general statement – It should be specific

A: This has been corected 

The conclusion is not in line with the aim of the study

A: This has been corected 

Introduction:

Line 45- Notable to understand what the authors wanted to convey.

A: This has been corected 

Line 47 – define comprehensive motor fitness

A: This has been corected 

The introduction does not contain the proper background of the study. There are a number of sentences written without any coherence.

A: This has been corected

Line 72 – Why this sentence in the introduction section.

A: This has been corrected

What is the rationale of the study? Why did the authors want to compare two sports?

A: This has been corrected

Methodology:

How was the sample size calculated?

A: This has been corrected

Add ethical approval details.

A: This has been corrected

Line 105 and 107 – was there any training session for the participants

A: The competitors were instructed in detail in the execution of the test. However, they were not given special training to avoid the effect of prior learning

How the participant was selected? How many volunteered? How many were excluded. Include a flow diagram

A: Added figure

Line 119 – provide details of the instrument

A: This has been corrected

The procedure has to be written in detail including the procedure for each test

Reliability and validity of the tests should be added to the methodology

A: This has benn corrected, tests are verified in terms of accuracy and reliability (test results can be found in the footnotes) 

Li e 187 – cause-effect cannot be established in a cross-sectional study

A: This has been corrected

The correlation coefficient is not available in results and table

Where is the training experience data?

A: This has been corrected

Discussion

Line 202 – is this the aim of the study?

A: This has been corrected

One of the most important part of your study is the correlation between training experience and fitness profile. This part is not discussed in the discussion section. Only one small paragraph with some results is available in the discussion regarding this. Even training experience data is not available in the results

A: This has been corrected

Line 280 – where is the correlation coefficient?

A: This has been corrected

Line 293- how you can conclude with this statement

A: This has been corrected

Line 297- wrong statement

A: This has been corrected

Conclusion not in line with the aim 0f the study 

A: This has been corrected

Round 2

Reviewer 1 Report

The authors have made significant changes to the manuscript and have answered all my questions. I have no further comments.

Author Response

Dear Reviewer, 

Thank you for your review. 

Your Sincerly, 

Łukasz Rydzik 

Reviewer 2 Report

Even though the authors tried to improve the paper there is a number of shortcomings still there in the manuscript. The authors didn’t follow most of the recommendations.

The introduction does not contain the proper background of the study.   

The rationale of the study is not written properly.

The methodology is poorly written.

 How the participant was selected? How many volunteered? How many were excluded? A participant’s flow diagram was recommended in the previous review.

Reliability and validity of the tests should be added to the methodology (which was recommended in the previous review).

Cause-effect cannot be established in a cross-sectional study

Author Response

Dear Reviewer,

Thank you very much for your time and valuable comments, which all have been considered and incorporated. The detailed list of responses is given below. We hope that the modifications and explanation will be acceptable for you.

Yours sincerely,

Rydzik, corresponding author

Even though the authors tried to improve the paper there is a number of shortcomings still there in the manuscript. The authors didn’t follow most of the recommendations.

A: Dear reviewer, thank you for your review, we tried our best to verify your comments. We hope that this time they will be acceptable to you.

The introduction does not contain the proper background of the study.   

A: Dear reviewer, we have not revised the title after the recent modifications. The title has now been corrected and the introduction has been improved 

The rationale of the study is not written properly.

A: This has been corrected 

The methodology is poorly written.

A: This has been corrected 

 How the participant was selected? How many volunteered? How many were excluded? A participant’s flow diagram was recommended in the previous review.

A: Added information and diagram

Reliability and validity of the tests should be added to the methodology (which was recommended in the previous review).

A: All tests have been checked for relevance and reliability, which is confirmed by references to the literature

Cause-effect cannot be established in a cross-sectional study

A: The aim of our study was, to analyse the fitness profile, to compare two different martial styles and to study the relationship, we did not draw conclusions related to the determination of the training effect. The conclusions are consistent with the nature of the research